# Multifaceted Chromatin Structure and Transcription Changes in Plant Stress Response

**DOI:** 10.3390/ijms22042013

**Published:** 2021-02-18

**Authors:** Jin-Hong Kim

**Affiliations:** 1Advanced Radiation Technology Institute, Korea Atomic Energy Research Institute, 29 Geumgu-gil, Jeongeup-si, Jeollabuk-do 56212, Korea; jhongkim@kaeri.re.kr; Tel.: +82-63-570-3333; 2Department of Radiation Science and Technology, University of Science and Technology, 217 Gajeong-ro, Yuseong-gu, Daejeon 34113, Korea

**Keywords:** chromatin, chromatin remodeling, histone modification, transcription, transcriptional reprogramming, plant stress response, DNA methylation

## Abstract

Sessile plants are exposed throughout their existence to environmental abiotic and biotic stress factors, such as cold, heat, salinity, drought, dehydration, submergence, waterlogging, and pathogen infection. Chromatin organization affects genome stability, and its dynamics are crucial in plant stress responses. Chromatin dynamics are epigenetically regulated and are required for stress-induced transcriptional regulation or reprogramming. Epigenetic regulators facilitate the phenotypic plasticity of development and the survival and reproduction of plants in unfavorable environments, and they are highly diversified, including histone and DNA modifiers, histone variants, chromatin remodelers, and regulatory non-coding RNAs. They contribute to chromatin modifications, remodeling and dynamics, and constitute a multilayered and multifaceted circuitry for sophisticated and robust epigenetic regulation of plant stress responses. However, this complicated epigenetic regulatory circuitry creates challenges for elucidating the common or differential roles of chromatin modifications for transcriptional regulation or reprogramming in different plant stress responses. Particularly, interacting chromatin modifications and heritable stress memories are difficult to identify in the aspect of chromatin-based epigenetic regulation of transcriptional reprogramming and memory. Therefore, this review discusses the recent updates from the three perspectives—stress specificity or dependence of transcriptional reprogramming, the interplay of chromatin modifications, and transcriptional stress memory in plants. This helps solidify our knowledge on chromatin-based transcriptional reprogramming for plant stress response and memory.

## 1. Introduction

The chromatin structure in eukaryotic cells contributes to genome stability but also displays dynamics to counter endogenous or exogenous stress factors, and neighboring chromatin structures may influence local nucleosome positions and gene expression [1,2,3,4]. Chromatin dynamics are epigenetically regulated by such things as histone variants, histone modifications, DNA (de-)methylation, nucleosome remodeling, and regulatory RNA, such as non-coding RNA (ncRNA), and are related to structural changes of nucleosomes or chromatin, which may affect genome-wide or locus-specific gene expression [5,6,7]. For example, the MEDIATOR OF PARAMUTATION1 (MOP1)-mediated RNA-directed DNA methylation (RdDM) activity regulates higher-order chromatin organization and genome-wide transcription in *Zea mays* [8]. The *Arabidopsis thaliana* SWITCH/SUCROSE NON-FERMENTABLE (SWI/SNF) family chromatin remodelers, BRAHMA (BRM) and SPLAYED (SYD) upregulate the expression of *CUP*-*SHAPED COTYLEDON* (*CUC*) genes to form and maintain boundary cells during embryogenesis [9]. Epigenetic modulators, modifiers, and mediators along with transcription factors and mediators concertedly coordinate nucleosome dynamics and gene expression, allowing alternative nucleosome configurations at transcription sites [10,11,12,13].

Plants have a distinctive three-dimensional spatial organization of chromatin, including similar but functionally different compartments, domains, and loop structures to those found in animals, likely as a result of the lack of topologically associated domains (TADs) or a CCCTC-binding factor (CTCF), which are required for transcriptional regulation in mammalian cells [14]. It is currently unclear as to why plants did not evolve these TADs or a CTCF, although they are equipped with a more sophisticated strategy for epigenetic regulation of gene expression at the nucleosome level involving diverse histone variants, post-translational histone modifications, and CG/CHG/CHH DNA methylation [14,15]. SWI/SNF chromatin remodelers affect the expression of underlying genes for various functions, including cell differentiation, organ development, flower morphogenesis, and phytohormone signaling during plant development by facilitating chromatin remodeling and dynamics [16]. H1 linker histones can also contribute to the transcriptional and epigenetic control of developmental and cellular transitions through fine-tuning of the spatial organization of euchromatin and heterochromatin [17]. Chromatin-based epigenetic regulation is flexible and dynamic to facilitate rapid adaptation to environmental changes; therefore, it can be crucial for dealing with environmental stress responses [1,2,18,19].

As sessile organisms, plants are inevitably exposed to various environmental stress factors, including cold, heat, salinity, drought, dehydration, submergence, waterlogging, and pathogen infection, throughout their lifetimes. To counter these, they are able to develop elaborated structural changes in chromatin and coordinate alterations in histone modification and DNA methylation for the expression of stress-responsive genes [1,2]. Reactive oxygen species (ROS) and nitric oxide (NO), which are indicators of stress responses, are considered pivotal in mediating chromatin dynamics through redox-dependent histone and DNA modifications [3]. The causal relationship between dynamic changes of chromatin composition and organization and stress-induced transcriptional alterations is undetermined in many cases, but an increasing number of reports support a correlation between these changes in diverse plant stress responses [20,21]. For example, the acetylation of H3K9 and H3K14, mediated by the histone acetyltransferase GENERAL CONTROL NON-REPRESSED PROTEIN5 (GCN5), contributes to thermotolerance or salt tolerance by activating heat or salt stress-responsive genes, such as *HEAT SHOCK TRANSCRIPTION FACTOR A3* (*HSFA3*), *UV*-*HYPERSENSITIVE6* (*UVH6*), *CTL1*, *PGX3*, and *MYB54* [22,23]. Additionally, the bivalent H3K4me3 and H3K27me3 marks associated with active genes in response to cold stress represent enhanced chromatin accessibility, which may facilitate the access of regulatory proteins required for gene expression [24]. Recently, it has been reported that the HIGH EXPRESSION OF OSMOTICALLY RESPONSIVE GENE15 (HOS15)-mediated chromatin modifications are required for the recruitment of C-REPEAT BINDING FACTOR (CBF) transcription factors to *COLD RESPONSIVE* (*COR*) gene promoters [25]. The association of differentially methylated regions (DMRs) with many differentially expressed genes (DEGs) in bok choy (*Brassica rapa*) and rice (*Oryza sativa*) under heat, drought, or salinity stress, reveals the interaction between DNA methylation and gene expression in plant abiotic stress responses [26,27].

Chromatin dynamics are also critical for stress signaling, adaptation, and memory in plants through stress-induced transcriptional reprogramming. The proteolytic and non-proteolytic roles of the ubiquitin–proteasome system are associated with the regulation and coordination of plant responses to environmental signals, particularly in relation to the E3 ligases, which function as regulators in the jasmonic acid (JA), salicylic acid (SA), ethylene, and abscisic acid (ABA) signaling pathways that coordinate responses to multiple environmental stresses [28,29]. In contrast, histone acetylation and methylation correlate with biotic or abiotic stress memory, such as systemic acquired resistance and heat and dehydration stress priming [30,31]. Various chromatin-based epigenetic regulatory mechanisms are becoming more relevant in relation to stress adaptation and somatic, intergenerational, and transgenerational stress memory [32,33]. Although previous reviews have analyzed chromatin remodeling and epigenetic regulation in plant stress responses [1,2,13,18,21,31], there is a need to update the available information on chromatin-based regulation of stress-responsive transcription and stress adaptation and memory in plants.

## 2. Chromatin Modification and Epigenetic Regulation in Plants

Epigenetic regulators are remarkably diversified in plants, facilitating the phenotypic plasticity of plant development and the survival and reproduction in unfavorable environments (Figure 1) [2,34]. Interactions of such functions as histone modifications, DNA methylation, and ncRNAs constitute a multilayered and multifaceted circuitry for sophisticated and robust epigenetic regulation in plants, which is important for their unique developmental processes, sessile lifestyle, and evolutionary traces. Aspects of plant development that are relevant to epigenetic regulation are as follows: (1) epigenetically flexible and transmissible gametogenesis, (2) late-producing germ cells to allow for the transmission of epigenetic modifications acquired by meristem cells, (3) vegetative or clonal propagation to perpetuate mitotically transmissible epigenetic states, (4) somaclonal variation during somatic embryogenesis, (5) diffusible epigenetic signals through plasmodesmata to modify the epigenetic state of stem cells and gametes, and (6) high tolerance for polyploidization and endoreplication associated with significant genomic and epigenetic changes [34].

Chromatin modifications and dynamics are necessary for genome replication, transcriptional activity, and DNA repair. The maintenance and transition of chromatin states, such as active euchromatin and repressed heterochromatin in plants, can be modulated by a variety of epigenetic regulators as well as the replication, transcription, and repair machinery (Figure 1) [35]. The epigenetic regulators identified in plants are divided into four groups: (1) chromatin remodelers (four conserved families: INO80/SWR1, SWI/SNF, CHD, and IMITATION SWITCH (ISWI)), (2) histone-modifying enzymes (methyltransferase, demethylase, acetyltransferase, deacetylase, ubiquitylase, kinase, etc.) and chaperones, (3) DNA (de)-methylation enzymes (CG/CHG/CHH methyltransferase and demethylase) and regulators, and (4) ncRNAs (miRNA, small-interfering RNA (siRNA), diRNA, and lncRNA) [13]. A nucleosome, the fundamental subunit of chromatin, consists of a histone octamer (two H2A, two H2B, two H3, and two H4) and a 147 bp DNA; therefore, the epigenetic modifiers responsible for histone modifications and DNA (de)-methylation (groups 2 and 3) are the primary epigenetic regulators in chromatin organization and its adaptability to environmental cues. Histone modifying enzymes and chaperones are highly conserved among eukaryotes, whereas the non-CG methylation at CHG and CHH sites are specific to plant genomes due to the plant-specific CHROMOMETHYLASE and RdDM pathways, which associate with the inverse correlation between heterochromatin and genic DNA methylation [36,37]. Groups 1 and 4 contribute to chromatin remodeling and dynamics through interactions with histone modifications and DNA methylation [13,38]. Although chromatin remodelers are somewhat conserved, they differentiate among plants, fungi and animals [39], whereas ncRNAs are the most variable epigenetic regulators [38,40].

Therefore, the engineering of plant genomes to modulate plant stress responses requires an updated understanding of stress-induced transcriptions and stress memory in the context of epigenetic regulation, such as histone modifications and DNA methylation. The following sections review: (1) chromatin dynamics, especially histone modifications and DNA methylation, associated with abiotic/biotic stress-induced transcriptions in plants, (2) interplay of histone modifications, DNA methylation, and ncRNAs in stress-induced transcriptional reprogramming, and (3) chromatin-based transcriptional reprogramming for stress signaling and memory in plants.

## 3. Chromatin Dynamics Associated with Abiotic Stress-Induced Transcriptions in Plants

Transcriptionally active euchromatic regions or genes are associated with histone 3 lysine 4 (H3K4) trimethylation (H3K4me3), H3K9/H3K23/H3K29 acetylation (H3K9ac/H3K23ac/H3K29ac), and/or CG/CHG/CHH hypomethylation [41]. Changes in histone modification and DNA methylation trigger differential expressions of stress-responsive genes to environmental cues [2,18]. Although the causal relationship between chromatin dynamics and transcriptional changes is still unclear for plant stress responses, there is accumulating evidence revealing the concurrent chromatin modifications in transcriptional stress responses [20,21]. The acetylation of core histones H3 and H4 is associated with transcriptional activation and generally induces a permissive chromatin structure, whereas histone deacetylation is often correlated with closed chromatin and transcriptional repression [42]. This review summarizes the recent updates that associate diverse chromatin modifications with the stress-induced transcriptions in plants under the major abiotic stresses, including cold, heat, salinity, drought, and flood (Table 1).

Cold: Cold stress in plants can be divided into two types: chilling (0–15 °C) and freezing (below 0 °C). Stress responses during cold acclimation and vernalization are epigenetically regulated by histone modifications and DNA methylation [43]. The histone deacetylase inhibitor Trichostatin A and DNA methylation inhibitor 5-Aza-2′-Deoxycytidine alter the expression of cold-induced genes in *Arabidopsis thaliana,* enhancing its cold tolerance [44], and in response to cold stress, histone H3 acetylation-mediated chromatin remodeling is necessary for transcriptional activation of a cold-inducible rice gene, *OsDREB1b* [45]. The POWERDRESS (PWR)-HOS15-HD2C complex in *Arabidopsis* negatively regulates the expression of *COR* genes through H3 deacetylation and repressive chromatin structure [25,46], whereas the histone deacetylase HD2C is degraded by the PWR-HOS15 complex, resulting in transcriptional activation of *COR* genes (*RD29A* and *COR15A*/*47*/*78*) under cold stress, by facilitating H3 acetylation and the permissive chromatin structure. In cold-treated banana (*Musa acuminata*) fruits, the increased levels of H3 and H4 acetylation in the promoters of ω-3 fatty acid desaturase genes (*MaFADs*) are correlated with the enhanced transcription of ω-3 *MaFADs*; the MaMYB4 transcription factor negatively regulates the transcription of ω-3 *MaFADs* by recruiting the histone deacetylase MaHDA2 [47]. The bivalent histone modifications H3K4me3 and H3K27me3 enhance chromatin accessibility and facilitate the access of regulatory proteins required for transcription of active genes in cold-stored potato (*Solanum tuberosum*) tubers [24]. The PHD (plant homeodomain) finger of the *Arabidopsis* SAP and MIZ1 domain-containing ligase1 (SIZ1), a small ubiquitin-like modifier (SUMO) E3 ligase, recognizes H3K4me3 and negatively regulates the histone lysine methyltransferase ATX1 for H3K4me3, associating the transcription and H3K4me3 levels of *WRKY70* under cold stress [48]. In addition, the differentially methylated *BramMDH1*, *BraKAT2*, *BraSHM4*, and *Bra4CL2* genes in cold-acclimated bok choy plant transcriptions demonstrate the importance of promoter demethylation [49]. The rice *OsOST1* (*Os03g0610900*), which participates in the ICE-CBF-COR pathway for cold tolerance, also associates its promoter demethylation with the increased gene expression [50]. In contrast, the cold-induced expression of *DREB1A* was repressed by the transgene-induced hypermethylation in the *DREB1A* promoter in the *Arabidopsis ice1-1* mutant [51]. The *Arabidopsis* ARGONAUTE1 (AGO1) promotes gene transcription by binding to the chromatin of stress-responsive genes through interaction with small RNAs and SWI/SNF chromatin remodeling complexes in response to cold or phytohormones [52].

Heat: Genetic and epigenetic controls through histone modifications, DNA methylation, chromatin remodeling, ncRNAs, and others are also involved in plant responses to heat [53]. The *Arabidopsis* histone acetyltransferase GCN5 mediates transcriptional activation of the heat stress-responsive *HSFA3* and *UVH6* genes by facilitating H3K9 and H3K14 acetylation in their promoter regions [23]. The histone chaperone ANTI-SILENCING FUNCTION1 (ASF1) A/B mediates H3K56 acetylation associated with the heat-induced activation of *HSFA2* and *HEAT SHOCK PROTEIN32* (*HSP32*) genes in *Arabidopsis thaliana* via nucleosome removal and RNA polymerase II (PolII) accumulation in the promoter and coding regions [54]. The *Arabidopsis* histone deacetylase HD2C interacts with a BRM-containing SWI/SNF chromatin remodeling complex through SWI3B, and represses the heat-activated *HSFA3* and *HSP101* genes by decreasing H4K16ac levels within their coding regions [55]. In contrast, the heat-induced transcription factor HSFA2 directly activates the H3K27me3 demethylase RELATIVE OF EARLY FLOWERING6 (REF6), which derepresses *HSFA2* to establish a heritable feedback loop generating transgenerational thermomemory in *Arabidopsis* [56]. After acclimatizing heat stress, the HSFA2 transcription factor directly associates with the promoter of the heat stress memory-related genes *APX2* and *HSP18.2* and facilitates the sustained accumulation of H3K4me2/3 at these gene loci as a transcriptional memory for hyper-induction of these genes during recurring heat stress [57,58]. Another H3K36 trimethylation, which is mediated by the histone methyltransferases SET DOMAIN GROUP8 (SDG8) and SDG26, is crucial for the processes of alternative splicing and flowering in *Arabidopsis* in response to fluctuating ambient temperature [59]. The *Arabidopsis* RdDM factor NRPD2, the second-largest subunit of PolIV and PolV, participates in transcriptional reprogramming of *At1g34220*, *At1g07590*, *At1g29475*, and auxin-responsive genes in response to temperature stress [60]. The *de novo* DNA methyltransferase *DRM2* and the largest subunits of PolIV (*NRPD1*) and PolV (*NRPE1*) genes, which are key players in the RdDM pathway, are upregulated in *Arabidopsis* under heat stress, and the heat-induced expression of *At3g50770* is associated with reduced promoter methylation [61]. In bok choy, heat-induced DMRs are generally located near the transcription start and end sites of gene-related regions and differentially correlate with gene expression, indicating CG/CHG/CHH context- and position-dependent transcriptional silencing [27]. During vegetative growth, the *Arabidopsis thaliana* imprinted gene *SDC* is suppressed by DNA methylation but is activated by heat stress, revealing its role in heat stress recovery [62]. In contrast, *Arabidopsis* FORGETTER1 (FGT1) binds to the proximal promoter of the heat-responsive *HSA32* and *HSP18.2*/*22.0* genes through nucleosome remodeling by interaction with SWI/SNF (BRM) and ISWI (CHR11/17) family chromatin remodelers and contributes to the sustainable induction of these genes after acclimatizing heat stress [63].

**Table 1 ijms-22-02013-t001:** Chromatin dynamics (histone modifications and DNA methylation changes) associated with abiotic stress-induced gene expression in plants.

Stress Type	Chromatin Modifications Tested	Epigenetic Modifiers Tested	Species	Genes Associated	Reference
Abiotic stress					
*Cold*	H3K9/14/27ac		*Oryza sativa*	*OsDREB1b*	[45]
	H3K9/14ac	PWR-HOS15-HD2C	*Arabidopsis thaliana*	*RD29A*, *COR15A*/*47*/*78*	[23,46]
	H3/H4ac	MYB4, HDA2	*Musa acuminata*	*MaFADs*	[47]
	H3K4/27me3		*Solanum tuberosum*	Cold-responsive genes	[24]
	H3K4me3	SIZ1, ATX1	*Arabidopsis thaliana*	*WRKY70*	[48]
	5-mC in promoter		*Brassica rapa*	*BramMDH1*, *BraKAT2*, *BraSHM4*, *Bra4CL2*	[49]
	5-mC in promoter		*Oryza sativa*	*OsOST1* (*Os03g0610900*)	[50]
	5-mC in promoter		*Arabidopsis thaliana*	*DREB1A*	[51]
	Chromatin remodeling	AGO1	*Arabidopsis thaliana*	Stimuli-responsive genes	[52]
*Heat*	H3K9/14ac	GCN5	*Arabidopsis thaliana*	*HSFA3*, *UVH6*	[23]
	H3K56ac	ASF1	*Arabidopsis thaliana*	*HSFA2*, *HSP32*	[54]
	H3K16ac	HD2C, BRM	*Arabidopsis thaliana*	*HSFA3*, *HSP101*	[55]
	H3K27me3	REF6, HSFA2	*Arabidopsis thaliana*	*HSFA2*	[56]
	H3K4me2/3	HSFA2	*Arabidopsis thaliana*	*APX2* and *HSP18.2*	[57,58]
	H3K36me3	SDG8/26	*Arabidopsis thaliana*	Genes subjected to alternative splicing	[59]
	5-mC	NRPD2	*Arabidopsis thaliana*	*At1g34220*, *At1g07590*, *At1g29475*, auxin-responsive genes	[60]
	5-mC in promoter	DRM2, NRPD1, NRPE1	*Arabidopsis thaliana*	*At3g50770*	[61]
	5-mC		*Arabidopsis thaliana*	*SDC*	[62]
	Chromatin remodeling	FGT1, BRM, CHR11/17	*Arabidopsis thaliana*	*HSA32*, *HSP18.2*/*22.0*	[63]
*Salinity*	5-mC, H3K9me2, H3K9ac		*Arabidopsis thaliana*	*SUVH2/5/8*, *ROS1*, *MSH6*, *APUM3*, *MOS6*, *DRB2*	[64]
	5-mC, H3K4me3, H3K9ac, H3K9me2		*Glycine max*	*Glyma11g02400*, *Glyma08g41450*, *Glyma20g30840*	[65]
	H3K9ac		*Zea mays*	*ZmEXPB2*, *ZmXET1*	[66]
	H3K9/14ac	GCN5	*Arabidopsis thaliana*	*CTL1*, *PGX3*, *MYB54*	[22]
	H3K27/36/56ac, H4ac, H3K9me2	PWR-HOS15-HDA9	*Arabidopsis thaliana*	*ERF4*/*5*/*6*/*11*, *STZ*, *KIN2*	[67]
	H3ac	HDA1, IDS1, TPR1	*Oryza sativa*	*LEA1*, *SOS1*	[68]
	H3ac	HDA5/14/15/18/19	*Arabidopsis thaliana*	*ABI5*, *NCED4*, *GA20 × 7*, *LEA4*_*2*, *P5CS1*, *NAC016*/*019*	[69,70]
	H4R3sme2	SKB1	*Arabidopsis thaliana*	*FLC*, *HAB1*, *MEK1*, *MEKK1*, *MRK1*	[71]
	H3K4me2/3	JMJ15	*Arabidopsis thaliana*	*WRKY33*, *ERF6*	[72]
	H3K4me3		*Arabidopsis thaliana*	*P5CS1*	[73]
	H3K4/27me3		*Ricinus communis*	*RSM1*	[74]
	H3K4/27me3	AGO2	*Oryza sativa*	*BG3*	[75]
	H2Bub	HUB1/2, UBC1/2/3	*Arabidopsis thaliana*	*PTP1*, *DsPTP1*, *MKP1*, *IBR5*, *PHS1*	[76]
	5-mC in promoter	24-nt siRNAs	*Arabidopsis thaliana*	*AtMYB74*	[77]
	5-mC in coding regions		*Triticum aestivum*	*TaHKT2*;*1*/*2*;*3*	[78]
	5-mC in promoter		*Medicago truncatula*	*LEA*, *AP2*/*ERF*, *KAT3*, *bZIP*, *WRKY*, *NAC*	[79]
	5-mC	SUVH7-BAG4-MYB106	*Oryza sativa*	*OsHKT1*;*5*	[80]
*Drought* *or dehydration*	H3K9ac, H3K4me3	AREB1, ADA26-GCN5	*Arabidopsis thaliana*	*RD20*, *RD29a*, *AtGOLS2*, *ProDH*	[81]
	H3K9ac		*Populus trichocarpa*	*PtrNAC006*, *PtrNAC007*, *PtrNAC120*	[82]
	H3K27ac	HAT1	*Arabidopsis thaliana*	*AtAREB1*	[83]
	H3/H4ac	MYB96, HDA15	*Arabidopsis thaliana*	*ROP6*/*10*/*11*	[84]
	H3K9ac	HDA9	*Arabidopsis thaliana*	Water deprivation-responsive genes	[85]
	Histone acetylation	HDA9	*Arabidopsis thaliana*	*CYP707A1*/*2*	[86,87]
	H3K4me3, H3K9me2		*Hordeum vulgare*	*HSP17*	[88]
	H3K4me3		*Arabidopsis thaliana*	*LTP3*, *LTP4*, *HIPP2.2*	[89]
	H3K4me3	JMJ17	*Arabidopsis thaliana*	*OST1*, *ABF3*, *ATHB7*, *ERD1*	[90]
*Submergence* *or waterlogging* *or others*	H3K4me3, H3ac		*Oryza sativa*	*ADH1*, *PDC1*	[91]
	H3/H4ac		*Triticum aestivum*	*CEL*, *PG*, *XET*	[92]
	5-mC, H3K27me3		*Arabidopsis thaliana*	*ASK11*, *CYP71A13*, *DREB19*, *AOX1D*, *At1g68620*, *ACS2*	[93]
	5-mC		*Sesamum indicum*	Drought and waterlogging-responsive genes	[94]
	5-mC	AGO1	*Arabidopsis thaliana*	*HR4*	[95]
	5-mC		*Oryza sativa*	Cd-responsive genes	[96]
	H3K4me3, H3ac		*Arabidopsis thaliana*	*CIPK11*, *RPA1E*, *GMI1*, *RAD51*, *AGO2*	[97]
	H3K14ac	C/S1-bZIP-SnRK1	*Arabidopsis thaliana*	*ETFQO*	[98]

Salinity: Salinity and drought stress also induces histone modification and DNA methylation changes, which modify the chromatin organization and dynamics to facilitate locus-specific gene expression in the genome in plants [99]. In the progeny of salt-stressed *Arabidopsis* plants, the lower expression levels of *SUVH2*/*5*/*8*, *ROS1*, *MSH6*, *APUM3*, *MOS6*, and *DRB2* genes can be associated with DNA hypermethylation, H3K9me2 enhancement, and/or H3K9ac depletion in the promoter or coding regions [64]. The soybean (*Glycine max*) genes *Glyma11g02400*, *Glyma08g41450*, and *Glyma20g30840*, are regulated by increased H3K4me3 and H3K9ac, reduced H3K9me2 as well as DNA hypermethylation [65]. Root swelling in maize (*Zea mays*) during salinity stress is related to the up-regulation of the cell wall-related genes *ZmEXPB2* and *ZmXET1*, associating with the elevated H3K9ac levels in the promoter and coding regions [66]. The *Arabidopsis* and wheat (*Triticum aestivum*) histone acetyltransferases AtGCN5 and TaGCN5 contribute to the expression of cellulose synthesis genes *CTL1*, *PGX3*, and *MYB54*, which are also associated with cell wall integrity and salt tolerance, by facilitating H3K9 and H3K14 acetylation [22]. In contrast, HDA9 constitutes a core histone deacetylase complex with PWR and HOS15, that binds to and directly represses many abiotic/biotic stress-responsive genes, including ethylene response factor (ERF) (*ERF4*/*5*/*6*/*11*), salt tolerance zinc finger (*STZ*), and kinase 2 (*KIN2*) genes, by modulating both histone acetylation (H3K27ac/H3K36ac/H3K56ac, H3.3K27/36ac and H4ac) and methylation (H3K9me2 and H3.1K36me2) [67]. Another histone deacetylase HDA1 induces the transcriptional repression of *LATE EMBRYOGENESIS ABUNDANT PROTEIN1* (*LEA1*) and *SALT OVERLY SENSITIVE1* (*SOS1*) in rice by decreasing H3 acetylation and interacting with the apetala2/ERF transcriptional repressors INDETERMINATE SPIKELET1 (IDS1) and TOPLESS-RELATED1 (TPR1) in the promoter regions of *LEA1* and *SOS1* genes, which are crucial for salt tolerance in rice [68]. The *Arabidopsis* class I (HDA19) and class II (HDA5/14/15/18) REDUCED POTASSIUM DEPENDENCY3 (RPD3)-like family histone deacetylases control positive and negative salinity stress responses, respectively, by acting conversely in the expressions of salt-responsive genes, such as *ABI5*, *NCED4*, *GA20x7*, *LEA4*_*2*, *P5CS1*, and *NAC016*/*019* [69,70]. In addition, the enhanced gene expression of six histone deacetylases HcHDA2, HcHDA6, HcHDA8, HcHDA9, HcHDA19, and HcSRT2 in kenaf (*Hibiscus cannabinus*) during salinity stress is associated with the levels of H3K9ac, H3K27ac, and H4K5ac, but has not yet been linked to other stress-induced transcriptional responses [100]. The *Arabidopsis* floral initiator Shk1 kinase binding protein1 (SKB1), a type II arginine methyltransferase, regulates transcription and pre-mRNA splicing of *FLOWERING LOCUS C* (*FLC*) and the stress-responsive genes, such as *HAB1*, *MEK1*, *MEKK1*, and *MRK1*, during salinity stress by dissociating from chromatin and altering the methylation status of histone 4 arginine 3 (H4R3) symmetric demethylation (H4R3sme2) and small nuclear ribonucleoprotein Sm-like4 (LSM4) [71]. Over-expression of the *Arabidopsis* H3K4 demethylase *JMJ15* gene down-regulates genes of stress-responsive transcription regulators, such as *WRKY33* and *ERF6*, which are marked with H3K4me2/3, thereby enhancing salt tolerance by modulating the expression of salinity-responsive genes, such as *RD29A*, *RD22*, and *COR15* [72]. In contrast, the salinity-induced transcriptional memory of Δ*^1^*-*pyrroline*-*5*-*carboxylate synthetase1* (*P5CS1*) for proline accumulation under chronic salinity stress is directly associated with light-dependent HY5 binding and the increased H3K4me3 level at the *P5CS1* promoter [73]. The bivalent H3K4me3 and H3K27me3 modifications contribute to the transcriptional regulation of the key salinity-response regulator RADIALIS-LIKE SANT (RSM1), an MYB-related transcription factor involved in salt stress signaling in castor bean (*Ricinus communis*) [74]. Salinity stress in rice causes the AGO2 proteins to be enriched on the *BIG GRAINS3* (*BG3*) locus under salinity stress and promotes the *BG3* expression by increasing H3K4me3 and decreasing H3K27me3 [75]. The *Arabidopsis* histone H2B monoubiquitination (H2Bub) mediated by two RING E3 ligases HUB1/2 and three E2 conjugases UBC1/2/3, regulates the expression of *PROTEIN TYROSINE PHOSPHATASE1* (*PTP1*) and *MAP KINASE PHOSPHATASE* (*MKP*) genes, including *DsPTP1*, *MKP1*, *IBR5*, and *PHS1*, required for salinity-induced microtubule depolymerization, and affects the activation of mitogen-activated protein kinase3 (MPK3) and MPK6 in salinity stress [76]. The salt-induced transcription factor AtMYB74, is transcriptionally regulated by the reduced levels of 24-nt siRNAs and RdDM in the promoter in *Arabidopsis* during stress conditions [77]. Salinity stress modulates the cytosine methylation and gene expression patterns in crop plants, including rice, barley (*Hordeum vulgare*), wheat (*Triticum aestivum*), and olive (*Olea europaea*), depending on the genotypes and tissue types [26,78,101,102,103]. Cytosine methylation in wheat (*Triticum aestivum*), which is induced by excess salinity, downregulates the expression of *Triticum aestivum HIGH*-*AFFINITY POTASSIUM TRANSPORTER2*;*1* (*TaHKT2*;*1*) and *TaHKT2*;*3* in the shoot and root tissues of the salt-tolerant genotypes, whereas *TaHKT1;4* is not affected [78]. In addition, the expression levels of salinity-responsive genes, such as *LEA*, *AP2*/*ERF*, *KAT3*, *bZIP*, *WRKY*, and *NAC* in caliph medic (*Medicago truncatula*) are somewhat associated with the altered cytosine methylation levels in the promoter regions [79]. The rice DNA methylation reader (OsSUVH7)-chaperone regulator (OsBAG4)-transcription factor (OsMYB106) complex activates the expression of *OsHKT1*;*5* under salinity stress by binding to the MYB binding *cis*-element (MYBE) as well as the miniature inverted-repeat transposable element (MITE) upstream of the MYBE in the *OsHKT1*;*5* promoter [80].

Drought or dehydration: Dynamic modifications of chromatin composition and organization also influence the regulation of transcription and stress responses in plants during drought stress and recovery [28,99,104]. In maize, variations in the euchromatic marks H3K9ac and H3K4me3 are associated with stress-induced gene expression in response to drought and recovery [104]. Chromatin dynamics through H3K9ac and H3K4me3 are correlated with the transcriptional activation and repression of the drought-inducible *RD20*, *RD29a*, *AtGOLS2* and the rehydration-inducible *ProDH* genes in *Arabidopsis* during drought stress and subsequent recovery [81]. The ISWI-responsive element (ABRE) binding protein1 (AREB1), which binds to ABRE motifs in promoters of the drought-responsive genes *PtrNAC006*/*007*/*120*, increases H3K9ac and PolII at these *PtrNAC* genes by recruiting the histone acetyltransferase unit ADA26-GCN5, which regulates drought response and tolerance in *Populus trichocarpa* [82]. Improved drought tolerance in *Arabidopsis* can be achieved by positively regulating the gene expression of AREB1 through H3K27ac enhancement at the promoter by the CRISPR/dCas9 system fused with histone acetyltransferase1 (HAT1) [83]. The *Arabidopsis* MYB96 transcription factor, which interacts and acts synergistically with the histone deacetylase HDA15 in ABA signaling, represses the expression of *RHO GTPASE OF PLANTS* (*ROP*) genes *ROP6*/*10*/*11* under drought stress by decreasing H3 and H4 acetylation at these promoters [84]. In contrast, the *Arabidopsis* histone deacetylase HDA9 negatively regulates plant responsiveness to both salinity and drought stresses by decreasing H3K9 acetylation levels in the promoters of many responsive genes [85]. The histone deacetylase HDA9 interacts with the ABA INSENSITIVE (ABI4) transcription factor and binds to the promoters of the ABA catabolism-related genes *CYP707A1*/*2* to repress the expression of *CYP707s* through histone deacetylation under drought stress [86,87]. Increased H3K4me3 and reduced H3K9me2 modifications at the coding regions of *HSP17* are associated with the *HSP17* expression in drought-treated barley plants [88]. When the stress-responsive genes are induced or repressed during dehydration stress in *Arabidopsis*, the H3K4me3 level at these genes is substantially altered compared to the moderate changes in the H3K4me1 and H3K4me2 levels [105]. The dehydration stress memory genes *LTP3*, *LTP4*, and *HIPP2.2*, which are super-induced during the recurrent dehydration stress in *Arabidopsis*, are also associated with the increased H3K4me3 and PolII levels at these genes during recovery from the initial stress [89]. The elevated H3K27me3 levels, which are mediated by the methyltransferase CLF and repress the flower-specific gene *AGAMOUS* (*AG)* at the vegetative stage, do not interfere with the transition to active transcription or with H3K4me3 accumulation during dehydration stress memory. The *Arabidopsis* histone demethylase JMJ17, which belongs to the KDMS/JARID12 family and demethylates H3K4me1/2/3, negatively regulates the stress-responsive genes *OST1*, *ABF3*, *ATHB7*, and *ERD1* under dehydration stress by demethylating H3K4me3 at those gene loci [90]. Drought-induced genome-wide DNA methylation changes are associated with differential transcription between contrasting rice genotypes in drought stress responses, depending on the genotype, development, and tissue specificity [26,106,107]. DNA methylation alterations during drought stress are restored to near-normal states during recovery in rice, *Eucalyptus globulus*, *Citrus sinensis*, and cotton (*Gossypium hirsutum*), and these methylation variations are found in many hormone-related genes associated with the drought stress response [106,108,109,110]. The genome-wide DNA methylation in *Arabidopsis* reacts strongly to water deprivation and the predominant changes are located at loci in the promoters of drought-responsive genes [111]. However, there are insufficient data to demonstrate a causal relationship between the gene- or locus-specific DNA methylation and the transcriptional changes of drought-responsive genes. Drought-associated epialleles in the DNA methylome are negligible and cannot be correlated with the drought-responsive gene expression under transgenerational drought stress, even in *Arabidopsis* [112].

Submergence or waterlogging or others: The diametric environmental water stresses drought and flood extremely affect water availability in plants, threatening food security, yet flood stresses, including submergence and waterlogging, are relatively understudied at the molecular level [113]. Since chromatin modifications and epigenetic regulation in water stress responses have been explored mainly under drought or dehydration conditions, there are limited available data to associate the chromatin-based transcriptional regulation with submergence or waterlogging stress responses [114,115]. One of the few studies revealed that the rice submergence-inducible *ADH1* and *PDC1* genes are activated during submergence by dynamic and reversible H3K4 trimethylation and H3 acetylation and increased binding of PolII at these genes [91]. Histone lysine butyrylation and crotonylation, along with acetylation, are dynamically regulated as a putative epigenetic mechanism for refining gene expression in rice under submergence [116]. The H3 and H4 acetylation in wheat seminal roots under waterlogging is positively associated with the increased transcription levels of the endoglucase (*CEL*), polygalacturonase (*PG*), and xyloglucan endotransglucosylase (*XET*) genes, which affect cell wall degradation and aerenchyma formation [92]. The age-regulated genes *ASK11*, *CYP71A13*, *DREB19*, *AOX1D*, *At1g68620*, and *ACS2*, which are induced under submergence by the ROS-activated transcription factor ANAC017, are more highly expressed in juvenile *Arabidopsis* plants than in adult ones, with lower promoter methylation and H3K27 trimethylation levels [93]. In sesame (*Sesamum indicum*), a drought-tolerant crop sensitive to waterlogging, drought stress strongly induces *de novo* DNA methylation, whereas flood stress decreases the level of DNA methylation, allowing a strong correlation between the transcript levels and methylation patterns of responsive genes [94]. The AGO1-dependent repression of the *HOMOLOG OF RPW8 4* (*HR4*) gene in *Arabidopsis* under submergence-induced hypoxia is associated with the AGO4-dependent RdDM pathway [95]. In addition, there is much evidence available to demonstrate the chromatin-based regulation of gene expression under other abiotic stresses. Heavy metal stress to cadmium (Cd) in rice causes transcriptional alteration of most of the DNA methylation-modified genes, including *GSH2*, *GSH35*, *LOX*, *HO1*, *OsIRO2*, *OsZIP1*, *HMT*, *CTF*, *Os09g02214,* and *OsSPL1* [96]. The *Arabidopsis* DNA damage response genes *CIPK11*, *RPA1E*, *GMI1*, *RAD51*, and *AGO2* are transcriptionally activated by the increased H3K4me3 and/or H3K9ac levels under genotoxic stress [97]. The *ELECTRON-TRANSFER FLAVOPROTEIN:UBIQUINONE OXIDO-REDUCTASE* (*ETFQO*) gene involved in branched-chain amino acid catabolism, an alternative mitochondrial respiratory pathway under carbohydrate-limiting conditions, is positively regulated in *Arabidopsis* during dark-induced starvation by the C/S1-bZIP-SnRK1 complex-facilitated enhancement of H3K14ac in the *ETFQO* promoter [98]. Overall, it is evident that chromatin modifications and epigenetic regulation are crucial to plant DNA damage responses, which are imposed by abiotic and biotic stress factors [13].

## 4. Chromatin Dynamics Associated with Biotic Stress-Induced Transcriptions in Plants

The main environmental biotic threats to plants include herbivores and microbial pathogens, and the latter is deservedly the primary biotic stress factor. Histone modifications H3/H4 acetylation, H3K4/H3K9/H3K27 methylation, H2B ubiquitination, DNA methylation, and chromatin remodeling are the main chromatin-based epigenetic mechanisms regulating the chromatin dynamics required for transcriptional activation and repression in plant biotic stress responses, such as plant-microbe interactions and plant immunity [19,117,118,119,120]. For example, H3/H4 lysine methylation and acetylation are associated with the transcriptional activation of the defense genes *WRKY6*/*29*/*53* in *Arabidopsis* leaves with inoculation of *Pseudomonas syringae* pv. *maculicola* (*Psm*) [30]. The gene expression of *PATHOGENESIS*-*RELATED GENE1* (*PR1*) in *Arabidopsis* upon pathogen infection is positively regulated by the SA-mediated H3/H4ac and H3K4me2/3 enhancement associated with diverse transcription factors at the *PR1* promoter [121]. SA is the major plant defense hormone against biotrophic and hemibiotrophic pathogens. Chromatin dynamics through H3K27me3 enrichment also contributes to transcriptional derepression for the spatiotemporal expression of virulence genes in the fungal wheat pathogen *Zymoseptoria tritici* after the initial stages of infection [122]. However, this review highlights the recent updates on chromatin-based transcriptional regulation in plant defense responses to pathogens, as opposed to virulence development of pathogens (Table 2).

The *Arabidopsis* elongator complex subunit2 (ELP2) and ELP3, which are important for both basal immunity and effector-triggered immunity (ETI), but are not involved in systemic acquired resistance (SAR), positively regulate the transcriptional induction of defense genes *PR1*/*2*/*5* through dynamic changes in histone acetylation and DNA methylation [123]. The histone acetyltransferase GCN5 is the catalytic subunit of the nucleosome-acetylating complex SPT-ADA2-GCN5 (SAGA). The cytoplasmic effect PsAvh23 of the soybean pathogen *Phytophthora sojae* suppresses the activation of defense-related genes *PGIP1*, *HSP20*/*90*, *WRKY33*/*41*, *NAC*, and *MAPKKK14*/*18* during infection by interfering with the association of ADA2 with GCN5 and decreasing H3K9 acetylation at these genes [124]. As a redox-signaling molecule, NO is induced by SA and affects histone acetylation by targeting and inhibiting histone deacetylase complexes, resulting in the hyperacetylation of specific genes, such as *MKK2*, *PNC2*, *BAG6*, *AIG2*, *TRX3*, *WRKY27/53*, and *TGA2*/*5* [125]. The effector HC-toxin (HCT) of the fungal pathogen *Cochliobolus carbonum* race 1 modulates both histone and non-histone protein acetylation by inhibiting histone deacetylase, also promoting virulence in maize [126]. The *Arabidopsis* histone deacetylase HD2B, which binds to and is phosphorylated by the MAP kinase MPK3 to modulate its subnuclear localization and target genes, contributes to the transcriptional repression of biotic stress response genes, including *At1g80180*, *At4g31470*, *At1g49640*, and *At5g01740* through H3K9 deacetylation in response to flagellin [127]. In contrast, the HDA6, an RPD3/HDA1-type deacetylase, acts as a general repressor of pathogen defense response by negatively regulating the histone acetylation and the expression of pathogen-responsive genes, including *PR1*/*2* and *WRKY38* in *Arabidopsis* with or without pathogen infection of the hemibiotrophic bacterial pathogen *Pseudomonas syringae* pv. *tomato* (*Pst*) DC3000 [128]. The *Arabidopsis* histone methyltransferases SDG8 and SDG25 participate in plant immunity at multiple levels, such as pattern-triggered immunity (PTI), ETI, and SAR in response to bacterial and fungal pathogens [129]. The SDG8 and SDG25 regulate the expression of *CAROTENOID ISOMERASE2* (*CCR2*) and *ECERIFERUM3* (*CER3*) genes, which are involved in the biosynthesis of carotenoid and cuticular wax, required for plant immunity, by inducing locus-specific H3K4/36 methylations and H2B ubiquitination at these genes. Another H3K4 methyltransferase, *Arabidopsis* Trithorax-Related7 (ATXR7), activates the expression of the pathogen-responsive *RECOGNITION OF PERONOSPORA PARASITICA4* (*RPP4*) and *SUPPRESSOR OF npr1*-*1 CONSTITUTIVE1* (*SNC1*) genes encoding nucleotide-binding leucine-rich repeat (NB-LRR) proteins through H3K4 methylation, in association with MOS9 [130]. The rice histone lysine demethylase JMJ704, a positive regulator of rice bacterial blight resistance, suppresses the transcription of the rice defense negative regulator genes, such as *NRR*, *OsWRKY62* and *Os*-*11N3,* by reducing the active H3K4me2/3 marks at these loci in response to *Xanthomonas oryzae pv. oryzae* [131]. Similarly, the *Arabidopsis* H3K9 histone demethylase JMJ27 negatively regulates the expression of *WRKY25*, a defense repressor gene, in response to *Pst* DC3000, but positively modulates the *PR1*/*3*/*4*/*5* transcription, both through H3K9 demethylation [132]. Another histone demethylase INCREASE IN BONSAI METHYLATION1 (IBM1) facilitates the transcriptional activation of defense marker genes *PR1*/*2* and *FRK1* in *Arabidopsis* with *Pst* DC3000 infection, allowing a permissive chromatin at these genes by increasing the active H3K4me3, and reducing the inactive H3K9me2 marks [133]. In addition, the rice BRHIS1, an SNF2 ATPase, which constitutes an SWI/SNF2 chromatin remodeling complex and is downregulated by the rice blast fungal pathogen *Magnaporthe oryzae*, suppresses the innate expression of defense-related *OsPBZc* and *OsSIRK1* genes through specific interaction with monoubiquitinated H2A.Xa/H2A.Xb/H2A.3 and H2B.7 variants in the absence of pathogen infection [134]. The *Arabidopsis* histone variant H2A.Z is monoubiquitinated by the Polycomb Repressive Complex1 (PRC1) components AtBMI1A/B/C, and this modification can be associated with transcriptional repression [135]. The chromatin-remodeling factor CHR5 in *Arabidopsis* contributes antagonistically to the upregulation of the immune receptor gene *SNC1*, with another chromatin remodeler DECREASED DNA METHYLATION1 (DDM1), and independently with a histone monoubiquitinase HUB1 [136].

As an epigenetic mark associated with chromatin dynamics in the genome, DNA methylation patterns are related to transcriptional regulation of genes as well as transposons and repetitive sequences in both host plants and plant pathogens [137,138]. Dynamic changes in DNA methylation of transposons or repetitive sequences can regulate not only their transcriptions but also those of neighboring genes in the SA-induced immune response of *Arabidopsis* [139]. Epigenetic quantitative trait loci (epiQTLs) in *Arabidopsis* are pericentromeric regions, and DNA hypomethylation in these loci contributes to the genome-wide priming of defense-related genes associated with quantitative disease resistance [140]. However, there are no *cis*-regulated defense genes in the resistance epiQTLs; therefore, the differentially methylated defense genes of *Arabidopsis* in response to the biotrophic pathogen *Hyaloperonospora arabidopsidis* (*Hpa*) are mainly *trans*-regulated by DNA (de)methylation [141]. In addition, the DMR-associated genes in watermelon (*Citrullus lanatus*) inoculated with *Cucumber green mottle mosaic virus* demonstrate no clear correlation between DNA methylation and gene expression [142]. By contrast, there are a few recent updates showing the relationship between the gene- or locus-specific DNA methylation patterns and the transcriptional levels of pathogen-responsive genes in plants. Dynamic DNA methylation and H3K9/14 acetylation changes induced by *Arabidopsis* ELP2 regulate the expression of *NONEXPRESSOR OF PATHOGENESIS*-*RELATED GENES1* (*NPR1*) and *PHYTOALEXIN DEFICIENT4* (*PAD4*) genes, which encode key regulators of plant immunity, during avirulent *Pst* infection [143]. DNA methylation is not directly linked with the expression of SA-induced defense genes *NPR1*, *StWRKY1*, or *PR1* in potatoes, whereas *R3a* promoter hypomethylation is required for its enhanced expression, contributing to the intergenerational defense priming to *Phytophthora infestans* (*P. infestans*) [144]. A cluster of disease resistance (R) genes encoding NB-LRR proteins can be regulated by the RdDM-mediated transcriptional silencing mechanism in the common bean (*Phaseolus vulgaris*) and rice [145,146]. In the wheat diploid progenitor, *Aegilops tauschii* infected by the biotrophic fungus *Blumeria graminis* f. sp. *tritici* (*Bgt*), the upregulation of a pathogenesis-related defense gene *AeGlu* is attributed to CHH hypomethylation in the promoter [147]. The increased transcription of *ENHANCED DISEASE SUSCEPTIBILITY1* (*EDS1*) in melatonin-treated grape (*Vitis vinifera*) berries with enhanced resistance to the necrotrophic fungus *Botrytis cinerea* (*B. cinerea*) also correlates with its decreased promoter methylation level [148].

## 5. Interplay of Histone Modifications, Chromatin Remodeling, DNA Methylation, ncRNAs, etc. in Stress-Induced Transcriptional Reprogramming

The concurrence of various chromatin or epigenetic modifications in plant abiotic/biotic stress responses suggests the importance of the cooperation of these epigenetic regulatory mechanisms in stress-induced transcriptional reprogramming. In particular, the simultaneous appearances of different histone modifications, such as acetylation, methylation, phosphorylation, and ubiquitination in plant stress responses, previously discussed in Section 4, are known to constitute some epigenetic regulatory networks [2,149]. The bivalent H3K4me3 (active) and H327me3 (repressive) marks at the same stress-responsive genes support the relevance of the interplay of different histone modifications in stress-induced transcriptional reprogramming [24,74]. In addition to the well-established interactions of histone acetylation, methylation, and ubiquitination shown in Table 1, the proportion of rice histone lysine acetylation and acylation, such as lysine butyrylation (Kbu) (H3K12/14bu and H2BK42/134bu) and crotonylation (Kcr) is dynamically and concertedly regulated in rice under submergence and starvation in order to adapt gene expression [116]. In contrast, the collaborative interactions between different types of epigenetic regulation, such as histone modifications, DNA methylation, chromatin remodeling, and ncRNAs, have been relatively less explored in plant stress responses [34,150], and their importance is only recently being examined in stress-induced transcriptional reprogramming. Therefore, this section discusses three types of epigenetic regulatory interplay in stress-induced transcriptional reprogramming: (1) histone modifications and chromatin remodeling, (2) DNA methylation and ncRNAs, and (3) histone modifications and DNA methylation (Figure 2).

Histone modifications and chromatin remodeling: Various chromatin-modifying proteins have been known to interact with histone modifiers or (in)active chromatin marks in transcriptional regulation or reprogramming [10]. The rice SWI/SNF2 ATPase BRHIS1 constitutes an SWI/SNF2 chromatin remodeling complex and regulates the expression of disease defense-related *OsPBZc* and *OsSIRK1* genes through specific interaction with monoubiquitinated H2A.Xa/H2A.Xb/H2A.3 and H2B.7 variants at those gene loci [134]. The *Arabidopsis* BRM chromatin remodeler represses the expression of heat-activated *HSFA3* and *HSP101* genes by removing H4K16ac at their chromatin loci through interaction with the HD2C [55]. In contrast, the *Arabidopsis* AGO1 binds to the chromatin of stress-responsive genes through interaction with small RNAs and SWI/SNF complexes and promotes their expression in response to cold and phytohormones [52]. Similarly, the rice AGO2 is associated with the enhanced H3K4me3 and the reduced H3K27me3 levels of the *BG3* gene to facilitate its expression under salinity stress through chromatin modifications [75]. The PWR proteins also constitute a chromatin-remodeling complex with either HOS15 and HD2C, or HDA9 and ABI4 to repress the cold-responsive *RD29A* and *COR15A*/*47*/*78* or the drought-responsive *CYP707A1/2* genes, respectively in *Arabidopsis* [46,87]. In addition, histone modifiers and transcription regulators together coordinate chromatin dynamics and gene expression, allowing alternative nucleosome configurations at transcription sites [10,11,12,13]. The human transcription factor TFIIH interacts with the histone acetyltransferase KAT2A on chromatin, and this partnership is crucial in the higher-order chromatin dynamics of gene expression [12]. In rice, the transcriptional repressors IDS1/TPR1 interact with the HDA1 to repress the salt-responsive *LEA1* and *SOS1* genes through H3 deacetylation at these gene loci [68]. The *Arabidopsis* MYB96 transcription factor and deacetylase HDA15 are interdependent in the regulation of *ROP* suppression through H3/H4 deacetylation under drought stress [84]. In contrast, the heat-induced HSFA2 transcription factor activates the H3K27me3 demethylase REF6 to regulate its expression through a heritable feedback loop for transgenerational thermomemory in *Arabidopsis* [56].

DNA methylation and ncRNAs: Small and long ncRNAs (sncRNAs and lncRNAs) play multidimensional roles in such areas as genome stability, chromatin organization, and (post-)transcriptional regulation [38,40]. Gene or locus-specific DNA methylation via the RdDM pathway occurs with the aid of sncRNAs, and there is a relationship between DNA methylation and histone modifications [151,152]. DNA methylation participates in transcriptional repression or activation, regulation of mRNA processing, silencing of transposons and repeats, and chromosome interactions [152]. Transposon-associated DMRs are coupled to the transcript abundance of nearby protein-coding genes in three rice cultivars with different drought and salt tolerances, and sncRNAs are positively correlated with hypermethylated regions, indicating interplay among DNA methylation, gene expression and small RNA abundance in rice abiotic stress response [26]. The reduced accumulation of 24-nt siRNAs in *Arabidopsis* under salinity stress is associated with transcriptional activation of the heavily methylated but salt-induced *AtMYB74* gene via the RdDM pathway [77]. The *DICER*-*LIKE4* (*DCL4*) gene promoter is hypomethylated in *Arabidopsis* with *Pst* infection, resulting in the increased expression of the long *DCL4* transcription start site isoform, which functions in a noncanonical siRNA pathway, producing a unique set of 21-nt siRNAs through the RdDM double-stranded RNA synthesis pathway [153]. In addition, the genome-wide DNA methylation patterns in soybean under continuous salinity stress support the combined epigenetic regulation by stress-associated lncRNAs and protein-coding genes [154].

Histone modifications and DNA methylation: The interplay between histone modifications and DNA methylation provides plants with a multifaceted and robust regulatory circuitry for transcriptional reprogramming in response to stress [34,152]. For example, DNA methylation changes as well as various histone modifications, such as H3K4me3, H3K9ac, and H3K9me2, are concertedly regulated for transcriptional activation or repression of the salt-responsive genes (*Glyma08g41450*, *Glyma11g02400*, *Glyma20g30840*, *SUVH2*/*5*/*8*, *ROS1*, *MSH6*, *APUM3*, *MOS6*, and *DRB2*) in soybean and *Arabidopsis* [64,65]. The rice transcriptional complex SUVH7-BAG4-MYB106, consisting of a DNA methylation reader, a chaperone regulator, and a transcription factor, activates the *OsHKT1*;*5* expressions during salinity stress [80]. In addition, the expression of the *Arabidopsis* pathogenesis-related gene *SNC1* is cooperatively regulated by the chromatin-remodeling proteins CHR5 and DDM1/SYD for nucleosome occupancy and DNA methylation, along with the histone modifiers HUB1/2 and ATXR7/MOS9 for H2Bub and H3K4me3, respectively, in plant immune responses [118,130,136]. However, the concerted interplays between different epigenetic regulators, including chromatin remodeling proteins and histone/DNA modifiers, are still elusive and remain to be investigated.

## 6. Chromatin-Based Transcriptional Reprogramming for Stress Signaling and Memory in Plants

Plants are sessile organisms that are unavoidably exposed to unfavorable environmental fluctuations over their lifetime and are able to fortify their defenses by adopting a multitude of epigenetic regulatory strategies for stress priming, somatic, intergenerational, and/or transgenerational stress memory [32,33,155]. Since stress memory can interfere with full recovery or cause adaptability concerns by sacrificing development and potential yield for a stronger or more rapid future defense response, the recovery process entails a balanced behavior between resetting and memory formation [156,157]. During recovery, RNA metabolism, posttranscriptional gene silencing (PTGS), and RdDM play key roles in resetting the epigenome and transcriptome and in altering memory. In addition, stress or defense signaling pathways are epigenetically controlled in plant stress responses through the dynamic engagement of mechanisms, such as histone modifications and DNA methylation [158,159]. This section discusses the recent updates in chromatin-based transcriptional reprogramming for stress signaling, priming, and stress memory in plants (Figure 3).

Stress signaling: Plant stress signaling pathways are concertedly regulated by multilayers of epigenetic regulators as well as transcription factors and phytohormones activated through the initial stress detection and are associated with the extensive transcriptional reprogramming of stress-related genes [158,159]. The transcription complex of IDS1-TPR1-HDA1 regulates salt stress signaling and tolerance in rice by mediating transcriptional repression of abiotic stress-responsive genes, including *LEA1* and *SOS1*, through direct interaction between two transcriptional repressors IDS1/TPR1 and a histone deacetylase HDA1 [68]. The *Arabidopsis* MYB96 transcription factor recruits the HDA15 to decrease H3/H4 acetylation at the *ROP* genes, which encode negative regulators of ABA signaling and represses these genes under drought stress [84]. The *Arabidopsis* AREB1, a key positive regulator of drought stress, can be transcriptionally activated by enhancing the H3K27ac at the promoter through the HAT1-fused CRISPR/dCas9 system to improve drought tolerance [83]. Histone H2B (de-)ubiquitination is a dynamic signaling path to chromatin-based control of plant stress responses by facilitating temporary shifts between transcriptionally permissive and repressive chromatin states during drought stress and recovery [28,29]. The *Arabidopsis* H2Bub regulates the expression of *PTP1* and *MKP1* genes for salinity-induced microtubule depolymerization and activates the PTP-MPK3/6 signaling module during salinity stress [76]. In addition, the *Arabidopsis* SWI/SNF chromatin remodeler SYD positively regulates the JA- and ethylene-responsive *PDF1.2a*, *MYC2*, and *VSP2* genes through direct interaction with PolII at those gene promoters in response to *B. cinerea* [160]. The *Arabidopsis* AGO1/4 regulates the expression of *HR4* via a noncanonical RdDM pathway in submergence-induced hypoxia signaling [95].

Stress priming or somatic stress memory: Priming can be defined as a transiently modified stress defense response to recurring stress, while somatic stress memory is mitotically heritable and lasts only for the remaining fraction of the lifespan of the organism [33]. RNA metabolism, PTGS, and RdDM play key roles in altering stress memory by resetting the epigenome and transcriptome during recovery [156]. In contrast, multiple layers of chromatin modifications, such as histone modifications, DNA methylation, chromatin remodeling, etc., contribute to plant stress responses, suggesting the role of chromatin dynamics in priming and somatic stress memory [31]. The histone H3/H4 lysine methylation and acetylation changes on the promoters of defense genes *WRKY6*/*29*/*53* act as transcriptional priming or memory for SAR in *Arabidopsis* with *Psm* inoculation [30]. Although the H3K9ac and H3K4me3 are enriched on the drought- or rehydration-inducible *RD20*, *RD29a*, *AtGOLS2*, and *ProDH* genes in *Arabidopsis* by drought stress, the former is rapidly reduced during the recovery by rehydration, but the latter is maintained as an epigenetic mark of stress memory [81]. The enriched H3K4me3 and H3K27me3 marks at the dehydration stress memory genes, such as *LTP3*, *LTP4*, and *HIPP2.2* during recovery from the initial dehydration stress, are associated with hyper-induction of these genes in *Arabidopsis* under the recurrent dehydration stress [89]. In addition, the H3K4me3 enhancement contributes to the salinity-induced transcriptional memory of *PSCS1* and proline accumulation upon repeated salinity stress in *Arabidopsis* [73]. The HSFA2-dependent sustained accumulation of H3K4me2/3 at the promoter of heat stress memory-related genes *APX2* and *HSP18.2* after acclimatizing heat stress functions as a transcriptional memory for hyper-induction of these genes during recurring heat stress [57,58]. The histone modifications H3K4me3/H3K9ac/H3K27me3 can be maintained or reset during stress recovery and serve as an epigenetic stress memory of drought-induced transcriptional variations during plant development in maize [104]. In contrast, cold acclimation alters DNA methylation patterns and confers heat tolerance in bok choy [49]. DNA methylation is not sufficient to increase heat-tolerance, but altered DNA methylation contributes to cross-adaptation. In addition, after acclimatizing heat stress in *Arabidopsis*, chromatin memory through nucleosome remodeling by the interaction of BRM/CHR11/17 chromatin remodelers and FGT1 is responsible for the sustainable induction of the heat-responsive genes *HSA32* and *HSP18.2*/*22.0* [63].

Inter- or transgenerational stress memory: Intergenerational memory is detectable only in the first stress-free generation, whereas transgenerational memory transcends at least two stress-free generations, depending on the epigenetic basis [33]. Contrary to the short-term, transient priming and somatic stress memory, the heritable inter- or transgenerational stress memory is challenging to define in the aspect of chromatin-based epigenetic regulation of transcriptional reprogramming and memory [161]. The highly dynamic and transient maternal hyperosmotic stress memory in *Arabidopsis* depends on the DNA methylation-associated transcriptional changes with an antagonistic relationship between the repressive epigenetic marks hypermethylated DMRs and decreased H3K27me3 marks to transmit the phenotypic plasticity to the immediate offspring [162]. The histone demethylase REF6, transcription factor HSFA2, and trans-acing siRNAs mediate the transcriptional reprogramming of *HEAT-INDUCED TAS1 TARGET5* (*HTT5*) for transgenerational thermomemory in *Arabidopsis* by constituting a coordinated epigenetic network [56]. In contrast, DNA methylation may be more relevant in the transgenerational defense memory of acclimated or primed plants. Although the *Arabidopsis* DNA methylome is stable and cannot be correlated with the drought-responsive gene expression under transgenerational drought stress [112], it responded globally to disease in previous generations and contributed to the transgenerational acquired resistance [163]. The promoter hypomethylation of the *R3a* resistance gene is required for its enhanced expression in descendants of the primed potato and is associated with intergenerational defense priming to *P. infestans* [144]. In addition, DNA hypomethylation patterns at some pericentromeric regions also contribute to the genome-wide priming of defense-related genes as heritable epigenetic marks, controlling quantitative disease resistance in *Arabidopsis* [140]. Transgene-induced promoter methylation can be associated with the heritable and transgenerational endogenous gene silencing [164]. Chromatin resetting mechanisms are critical for the prevention of unnecessarily stable transgenerational transmission of stress memory, as well as in balancing between resetting and memory formation during recovery [156,157].

## 7. Concluding Remarks and Perspectives

In this review, the latest studies show that chromatin-based transcriptional regulation is important for both immediate response and future memory in relation to plant stress response; however, multifaceted epigenetic regulation of stress response and memory in plants is still elusive. Different categories of stress memory genes associated with “transcriptional memory”, “epigenetic memory,” or “delayed memory” suggest distinctive roles of epigenetic signatures in stress-induced transcriptional reprogramming and memory [104]. Differential or cooperative interactions of diverse epigenetic regulators have been poorly studied, and the results or interpretations of these interactions are not cohesive. Increased practical approaches are being attempted for enhancing plant stress tolerance through chromatin-based epigenetic regulation of transcriptional reprogramming and memory. The CRISPR-dCas9 system fused with a histone acetyltransferase and/or a methyltransferase as well as a transcriptional activator can be used to facilitate transcriptional priming through chromatin modifications of specific target genes in plants [83,165]. Recently, the importance of chromatin dynamics and epigenetic modifications has become widely recognized in plant stress responses from model plants to crops [21,115,138,166], and some epigenetic regulators, such as histone and DNA modifiers, can be promising targets for increasing stress tolerance and yield in crop plants, including rice, wheat, and barley under abiotic and biotic stresses. In addition, transcriptional reprogramming and memory through chromatin-based epigenetic regulation is the basis of the stress response and memory and is key to improving crop productivity. Therefore, it is necessary to elucidate the interactions of the different epigenetic regulators or modifications and their regulatory mechanisms for transcriptional reprogramming and memory in plant stress response.

## Figures and Tables

**Figure 1 ijms-22-02013-f001:**
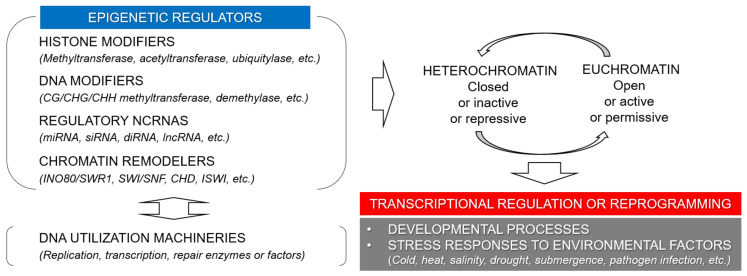
Epigenetic regulators mediate chromatin dynamics for transcriptional regulation or reprogramming in plant development and stress response. Histone and DNA modifiers, regulatory non-coding RNAs (ncRNAs), chromatin remodelers, and DNA utilization machineries together constitute a multilayered and multifaceted circuitry for chromatin-based transcriptional regulation in plant development and stress response.

**Figure 2 ijms-22-02013-f002:**
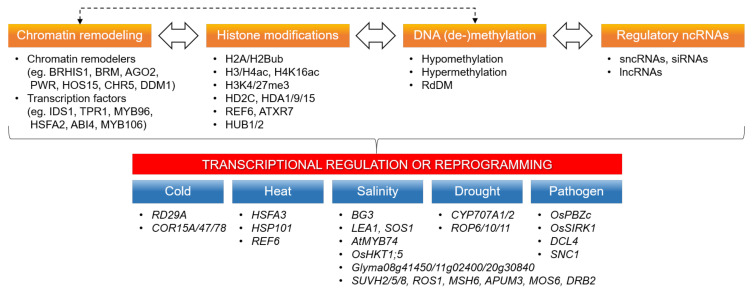
Interplay between histone modifications, chromatin remodeling, DNA (de-)methylation, and ncRNAs in stress-induced transcriptional reprogramming. Cooperative interactions of histone and DNA modifications, histone variants, chromatin remodelers, transcription factors, and ncRNAs in response to cold, heat, salinity, drought, and pathogen infection represent multilayered and multifaceted epigenetic regulation of stress-induced transcriptional reprogramming.

**Figure 3 ijms-22-02013-f003:**
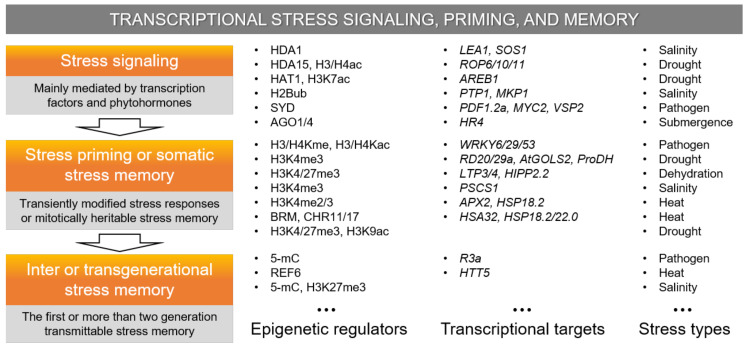
Chromatin-based transcriptional reprogramming for stress signaling, priming, and memory in plant stress response. Pivotal roles of epigenetic regulators or chromatin modifications in plant stress response are somewhat understood in association with transcriptional priming or memory but still remain unclear in heritable transcriptional reprogramming; therefore, they are to be further explored, particularly in relation to inter- or transgenerational stress memory.

**Table 2 ijms-22-02013-t002:** Chromatin dynamics (histone modifications and DNA methylation changes) associated with biotic stress-induced gene expression in plants.

Stress Type	Chromatin Modifications Tested	Epigenetic Modifiers Tested	Species	Genes Associated	Reference
Biotic stress					
	H3/H4ac, H3/H4me		*Arabidopsis thaliana*	*WRKY6*/*29*/*53*	[30]
	H3/H4ac, H3K4me2/3, H3K9me2		*Arabidopsis thaliana*	*PR1*	[121]
	Histone acetylation, 5-mC	ELP2/3	*Arabidopsis thaliana*	*PR1*/*2*/*5*	[123]
	H3K9ac	PsAvh23, ADA2, GCN5	*Glycine max*	*PGIP1*, *HSP20*/*90*, *WRKY33*/*41*, *NAC*, *MAPKKK14*/*18*	[124]
	H3K9/14ac	NO	*Arabidopsis thaliana*	*MKK2*, *PNC2*, *BAG6*, *AIG2*, *TRX3*, *WRKY27*/*53*, *TGA2*/*5*	[125]
	H3K9ac	HD2B	*Arabidopsis thaliana*	*At1g80180*, *At4g31470*, *At1g49640*, *At5g01740*	[127]
	H3ac	HDA6	*Arabidopsis thaliana*	*PR1*/*2*, *WRKY38*	[128]
	H3K4/36me1/2/3, H2Bub	SDG8/25	*Arabidopsis thaliana*	*CCR2*, *CER3*	[129]
	H3K4me3	ATXR7, MOS9	*Arabidopsis thaliana*	*RPP4*, *SNC1*	[130]
	H3K4me2/3	JMJ704	*Oryza sativa*	*NRR*, *OsWRKY62*, *Os*-*11N3*	[131]
	H3K9me2	JMJ27	*Arabidopsis thaliana*	*WRKY25*, *PR1*/*3*/*4*/*5*	[132]
	H3K4me3, H3K9me2	IBM1	*Arabidopsis thaliana*	*PR1*/*2*, *FRK1*	[133]
	H2A.Xa/H2A.Xb/H2A.3/H2B.7ub	BRHIS1	*Oryza sativa*	*OsPBZc*, *OsSIRK1*	[134]
	Chromatin remodeling	CHR5	*Arabidopsis thaliana*	*SNC1*	[136]
	5-mC	ELP2	*Arabidopsis thaliana*	*NPR1*, *PAD4*	[143]
	5-mC in promoter		*Solanum tuberosum*	*R3a*	[144]
	5-mC		*Phaseolus vulgaris*	*NB*-*LRR*	[145,146]
	5-mC		*Aegilops tauschii*	*AeGlu*	[147]
	5-mC in promoter		*Vitis vinifera*	*EDS1*	[148]

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
