# Peer review of "Multifaceted Chromatin Structure and Transcription Changes in Plant Stress Response"

_ijms, 2021, doi:10.3390/ijms22042013_

Round 1

Reviewer 1 Report

The objective of the review is to take stock of recent work concerning epigenetic modifications occurring in plants in response to stress. The review is primarily a compilation of data with some redundancy. It is, however, extremely comprehensive and exhaustive, and well written. I have very few comments to make.   Redefine the page number from page 11. line 102 : "Although previous reviews have ..." please cite them. line 131 :  replace  "four conserved families, INO80/SWR1 ..." by "four conserved families: INO80/SWR1 ..."   Table 1 : The organization of the table makes it unclear to know what type of stress is being referred to.   line 401: define what is "Cd stress"   line 404 : "The Arabidopsis DNA damage response genes CIPK11, RPA1E, GMI1, RAD51, and AGO2 are transcriptionally activated by the increased H3K4me3 and/or H3K9ac levels under genotoxic stress"
  • I  think that DNA damage response deserves a dedicated section given the abundant literature on the subject. Maybe you should just put few relevant references?

Author Response

Comments and Suggestions for Authors

The objective of the review is to take stock of recent work concerning epigenetic modifications occurring in plants in response to stress. The review is primarily a compilation of data with some redundancy. It is, however, extremely comprehensive and exhaustive, and well written. I have very few comments to make.

Redefine the page number from page 11.

=> The page numbers in the manuscript were serially redefined.

line 102: "Although previous reviews have ..." please cite them.

=> It was replaced with “Although previous reviews have analyzed chromatin remodeling and epigenetic regulation in plant stress responses [1, 2, 13, 18, 21, 31], …"

line 131:  replace "four conserved families, INO80/SWR1 ..." by "four conserved families: INO80/SWR1 ..."

=> Corrected as suggested.

Table 1: The organization of the table makes it unclear to know what type of stress is being referred to.

=> The original manuscript seems to be modified in the format and transferred to the reviewers. Since Table 1 contains much information, it can be clearer in a separate landscape page as in the original manuscript. In addition, horizontal lines were added to differentiate stress types.

line 401: define what is "Cd stress"

=> It was replaced with “Heavy metal stress to cadmium (Cd)…”

line 404: "The Arabidopsis DNA damage response genes CIPK11, RPA1E, GMI1, RAD51, and AGO2 are transcriptionally activated by the increased H3K4me3 and/or H3K9ac levels under genotoxic stress" I think that DNA damage response deserves a dedicated section given the abundant literature on the subject. Maybe you should just put few relevant references?

=> I understand the reviewer’s suggestion. However, heavy metal stress and genotoxic stress are just mentioned in the manuscript as additional abiotic stress types and these are not recognized by plant biologists as a major type of natural environmental stress. Since this review paper is long enough in this state, I would like to avoid unnecessary adding to this section. Also, chromatin modification and epigenetic regulation in plant DNA damage response were covered in my previous review [13].

Reviewer 2 Report

In this review, the manuscript author explains a vital topic that is the role of chromatin in plant stress response. The manuscript is very well written and covers all aspects of the stress response. Figures and tables are very well described, and the manuscript covers recent examples also. I found no significant faults in the manuscript hence recommending it for publication in its current format.

Minor:

At L44, it is written as upregulated; however, At L231, it is written as up-regulated. Please change either of them.

At L219 change from After an acclimatizing heat stress to After an acclimatizing heat stress.

At L339 ABA signaling, represses to ABA signaling represses.

At L356 at the vegetative stage, do not interfere to at the vegetative stage do not interfere.

At L528 stress responses suggests to stress responses suggest.

At L532 in the section 4 to in section 4.

At L598 siRNAs though the RdDM to siRNAs through the RdDM.

At L690 after an acclimatizing heat stress in Arabidopsis to after an acclimatizing heat stress in Arabidopsis.

Author Response

Comments and Suggestions for Authors

In this review, the manuscript author explains a vital topic that is the role of chromatin in plant stress response. The manuscript is very well written and covers all aspects of the stress response. Figures and tables are very well described, and the manuscript covers recent examples also. I found no significant faults in the manuscript hence recommending it for publication in its current format.

Minor:

At L44, it is written as upregulated; however, At L231, it is written as up-regulated. Please change either of them.

=> The word of “up-regulated” was replaced with “upregulated”.

At L219 change from After an acclimatizing heat stress to After an acclimatizing heat stress.

=> No difference.

At L339 ABA signaling, represses to ABA signaling represses.

=> Corrected as suggested.

At L356 at the vegetative stage, do not interfere to at the vegetative stage do not interfere.

=> Corrected as suggested.

At L528 stress responses suggests to stress responses suggest.

=> Since the subject of this sentence is “The concurrence of …”, the verb should be “suggests …”

At L532 in the section 4 to in section 4.

=> Corrected as suggested.

At L598 siRNAs though the RdDM to siRNAs through the RdDM.

=> Corrected as suggested.

At L690 after an acclimatizing heat stress in Arabidopsis to after an acclimatizing heat stress in Arabidopsis.

=> No difference